# An empirical estimation of aggregate import demand under foreign exchange constraints: Evidence from Ethiopia

**Mohammed Yimam Ali** [1]*, **Ahmed Mohammed Yimer**[2], **Tsadiku Setegn Dessie**[2]

**1** Department of Economics, Woldia University, Woldia, Ethiopia, **2** Department of Marketing Management, Woldia University, Woldia, Ethiopia

* muhammed.y@wldu.edu.et

**Data Availability Statement:** All relevant data are within the manuscript and its Supporting Information files.

**Funding:** The author(s) received no specific funding for this work.

## Abstract

This study employs the estimation of aggregate import demand under foreign exchange constraints in Ethiopia, utilizing annual time series data from 1985 to 2021. The regression analysis is conducted using the nonlinear autoregressive distributed lag (NARDL) approach to investigate the impact of the accumulation of foreign exchange reserves on aggregate import demand in Ethiopia. The estimation results indicate that, in the long run, all the variables, i.e., foreign exchange reserve, the relative price of imports, real income, volatility of the exchange rate, money supply, and policy dummy, significantly determine the behavior of aggregate imports over the reference period. The findings also show that, in the long run, foreign exchange reserve, real income, and the exchange rate positively affect the demand for imports in Ethiopia. Meanwhile, a positive shock in relative import price and money supply negatively affects import demand in Ethiopia. Thus, the price and income elasticity estimates have correct signs and are statistically significant. The variables included in the model strongly affect import demand in both the short and long run. Finally, policymakers aiming to significantly influence import demand should focus on effective management of these variables, as they strongly affect import volume.

## Introduction

In today's interconnected global landscape, mutual interdependence among countries through globalization is evident. The world increasingly behaves as though it were part of a single market, with interdependence in the production, consumption, and distribution of goods and services. This interdependence is reflected in the growth of world trade as a proportion of output, with the ratio of world imports to world gross domestic product alarmingly increasing from 7% in the 1930s to about 10% in the 1970s and over 18% in 1996 [1].

International conditions offer opportunities for both developed and developing countries, with less developed countries relying on developed nations for finance, technology, and technical workforce, while advanced countries depend on less developed ones, especially for raw materials [2]. International trade believed to create a favorable atmosphere that contributes to the economic growth and development of participating nations [3].

**Competing interests:** The authors have declared that no competing interests exist.

In the global context, consumers are accustomed to seeing goods and services from every corner of the globe in their retail shops and local wholesale markets. These foreign products (imports) provide more choices to users, as they usually manufactured at a lower opportunity cost than any domestically produced equivalent. Imports help consumers manage their household finances [4] Countries need foreign reserves to pay external debts, afford capital for sectors of the economy, and profit from diversified portfolios. An acute shortage of foreign exchange reserves is a major problem in developing countries, further limiting output growth and becoming a crucial challenge. The animated role played by imports in external trade and development has generated widespread interest in explaining the determinants of imports under foreign exchange constraints in developing countries [5] In most international economic literature, the problem usually discussed within the framework of the "two-gap" models developed. Based on this model, a foreign exchange shortage becomes an almost absolute constraint on the economic growth of a nation through domestic savings that are available in sufficient amounts.

Neo-classical growth theories emphasize the role of relative prices, particularly exchange rate adjustment, as a means for overcoming any foreign exchange constraint. The implication is that the foreign exchange gap reflects an overvalued real exchange rate. Nonetheless, if the exchange rate is determined by the force of demand and supply of foreign currency (i.e., a flexible exchange rate), there can be no foreign exchange gap [6].

Imports play a pivotal role in foreign trade and the economic development of a nation. Unfortunately, in Ethiopia, only scanty empirical evidence exists to explain import behavior.) [7] examined the Economic Implications of Foreign Exchange Rationing in Ethiopia, finding that due to the increase in foreign transfers and capital inflows, the country enjoyed remarkable economic growth from 2004/05 to 2008/09. However, this rapid growth was accompanied by a major appreciation of the real exchange rate, reducing incentives for domestic production of exportables and non-protected importables. In addition, major external shocks to the economy aggravated foreign exchange and macro-economic imbalances. They considered imports within a wider trade model, primarily concerned with the impact of the parallel foreign exchange market, without drawing specific import policy implications from their evidence. Policymakers have struggled to devise import strategies that promote growth without a significant deterioration in the trade balance, unable to predict the response of imports to external and domestic shocks in the presence of foreign exchange constraints, the import strategies have not achieved their desired goals.

Considering the Ethiopian economy is open to international trade and finance and heavily import-dependent, the behavior of imports has strong implications for the external balance. The foregoing background analysis of Ethiopia exposes the weakening impact of imports on the balance of payment (BOP), as evidenced by a persistent current account deficit of about -2.72 USD in 2020 [8].

Currently, in Ethiopia, the supply of foreign currency available for importers and travelers is increasingly facing chronic shortages, falling into a whirlpool, and the parallel market for hard foreign currency is growing in the country. As a result, net exports and terms of trade will deteriorate, debt service costs will increase, and this leads to the limited import capacity of the nation. Therefore, the motivations of this study are to explicitly examine the determinants of import demand under foreign exchange constraints in Ethiopia. There are five types of models regarding estimating import demand function for estimating aggregate import demand function:

1. The traditional model: This model incorporates relative income and price to determine import demand behavior, with income measured as real GDP.

2. The Senhadji model (the revised traditional model): Essentially, the revised model also uses relative price and income, with income measured as real GDP minus exports.

3. The disaggregated or decomposed GDP model: This approach decomposes GDP into three categories—final consumption expenditure, expenditure on investment goods, and exports [9]

4. The "National Cash Flow" model (the dynamic structural import demand model): The model, developed [10] takes into account a growing economy, rather than an endowment economy, and considers investment and government activity. The model replaces real GDP with a "national cash flow" variable.

5. The "Emran & Shilpi" model: This model employs the general import model to analyze the demand for import under foreign exchange constraints, incorporating a structural model that considers a binding foreign exchange constraint.

In this study, the researcher attempts to contribute to the existing body of literature by considering three main aspects. Firstly, the researcher considers the reserve tranche position in the International Monetary Fund (IMF) and Special Drawing Right (SDR) as part of foreign exchange receipts into the current import decision. Secondly, money supply and volatility of the exchange rate are included in the model, which are leading determinant factors for determining import demand. Moreover, the researcher updates the work by Emran and Shilpi models of import demand, with the inclusion of the latest data and the use of a representative agent model on the variables under consideration. Thirdly, since the specification issues in the import demand model are liable to bias and errors if import liberalization is not considered between the Derg and EPRDF regime, in this study, the researchers have looked at the effect of import liberalization, which is captured by the inclusion of a dummy variable.

Finally, this study adequately explains the determinants of aggregate official imports and components in Ethiopia, explicitly showing the role of foreign exchange reserves in influencing import demand. Estimating the import demand function is an important step in providing a foundation for rational, evidence-guided decision-making, partially filling the information gap, and aiding policymakers to predict the response of imports under foreign exchange constraints. The main objective of this study is to estimate aggregate import demand under foreign exchange constraints in Ethiopia from 1985 to the 2021 fiscal year.

## 2. Literature review

### 2.1 Import behavior in the Keynesian general theory

A well-known Keynesian import behavior theory elucidates the connection between import and economic growth, advocating for export-led growth over import-led growth. This theory, which focuses on the demand side of the economy, operates under the following assumptions:

- Assumption 1: Increases in aggregate demand drive economic growth toward the full employment level of output.

- Assumption 2: Components of aggregate demand (consumption, investment, government expenditure, and net exports) influence economic growth proportionally to the increase in aggregate expenditure multiplied by the related expenditure multiplier.

- Assumption 3: Net exports, equal to exports minus imports, positively impact national income.

- Assumption 4: There exists a negative relationship between imports and national income due to imports decreasing total demand, which is a proxy variable for economic growth.

- Assumption 5: Imports exhibit a negative multiplier effect on output, similar to savings, representing a leakage from aggregate demand.

   Furthermore, the role of relative import prices (proxied by the real exchange rate) is significant in determining import demand, based on the following assumption:

- Assumption 6: The traditional import demand function assumes an inverse relationship between the quantity of imports demanded and relative import prices, holding real income constant.

**2.1.1 Import behavior and trade policy.**   Much of the economic literature focusing on imports and international trade emphasizes trade policy. Scholars such as [11]) argue that trade policy boils down to protection and, therefore, government intervention, particularly in an LDC (Less Developed Country) context. This discussion operates under the assumption that:

- Assumption 7: Trade policies, while politically efficient, are economically inefficient and have substantial negative effects on economic growth.

**2.1.2 Income and price elasticities in trade theory.**   Various economic literatures acknowledge the importance of income and price elasticities in international trade due to their theoretical and policy implications [12]. In modern international trade theory, three major frameworks—comparative advantage theory, Keynesian trade multiplier theory, and new trade theory (imperfect competition theory of international trade) explain the roles of income and price in determining trade. These frameworks operate under the following assumptions:

- Assumption 8: The neoclassic trade theory of comparative advantage assumes fixed employment levels and production always on a given frontier.

- Assumption 9: The Keynesian framework assumes inelastic relative prices and varying employment levels.

- Assumption 10: The imperfect competition theory of trade considers economies of scale, product differentiation, and monopolistic competition in international trade.

**2.1.3 The Imperfect Substitutes Model.**   [4] introduce two critical trade models: imperfect substitutes and perfect substitutes, to empirically examine the price and income effects on foreign trade. The imperfect substitute's model mainly applied to study manufactured goods and aggregate imports, assuming that imports and exports are not perfect substitutes for domestic goods.

## 2.2 Empirical literature review

Numerous studies have explored the relationship between foreign exchange reserves and aggregate import demand, particularly in developing countries. These empirical investigations shed light on the determinants of import behavior and offer valuable insights into the economic dynamics at play.

   [13] analyze the import demand function for CIBS (China, India, Brazil, and South Africa) from 1970 to 2007. Their study, employing Auto-Regressive Distributed Lag (ARDL) models and bounds tests for cointegration, reveals higher long-run income elasticities for CIBS compared to earlier studies.

[14] investigates import and export demand equations for six developing countries using Johansen's cointegration approach from 1973 to 1990. The findings emphasize the influence of relative import prices, domestic income, and the nominal effective exchange rate on import volume, satisfying the Marshall-Lerner condition. [15] estimate the import demand function in developing countries, applying a structural econometric approach with a rational expectation hypothesis of a permanent income model. Stable parameter estimates are observed across countries, indicating the robustness of the model.

[9] examines the empirical evidence on the co-integration between aggregate import demand behaviors for five Asian countries. The study suggests a cointegration relationship between import volume, national cash flow, and relative price of imports in Malaysia and Singapore, with mixed results for other countries. [16] analyze aggregate import demand using panel data for Latin American and Caribbean countries from 1975 to 2005. Their study reveals a negative correlation between import demand and relative prices, while real income positively affects import demand.

These empirical studies contribute to a comprehensive understanding of import behavior by highlighting the role of factors such as income, relative prices, and exchange rates. They offer valuable insights into the determinants of import behavior in different countries, providing a foundation for further empirical research in this area.

## 3. Methodology

### 3.1 Theoretical framework and model specification

To explore the impact of foreign exchange reserves on aggregate import demand behavior in Ethiopia, our model draws on the analyses of) [15, 17, 18]. The theoretical framework underlying our model includes the following assumptions:

Assumptions:

1. Rational expectations: Agents form expectations about future variables based on all available information.

2. Permanent income model: Representative agents maximize utility by consuming both domestically produced (Dt) and imported goods (Mt) subject to a dynamic budget constraint.

3. Binding foreign exchange constraint: The optimization problem is constrained by the availability of foreign exchange reserves (Ft).

The optimization problem of the representative agent is as follows:

$$Max(Dt, Mt, At)V = E \int_{t=0}^{\infty} e^{-\delta t} U(Dt, Mt) dt$$

Subject to

$$\frac{dAt}{dt} = rAt + \hat{Y}t - Dt - PtMt \tag{1}$$

$$PtMt \leq Ft \tag{2}$$

Where, *Pt* as the relative price of imports at prevailed exchange rate, *At* as assets, *t* as labor income *Ft* as the amount of foreign exchange availability (reserve) and *r* as the constant real interest rate. δ as subjective rate of time preference in which the representative agents discount the future and Å $\left(\frac{dAt}{dt}\right)$ is a time derivative.

Accordingly, the current value Hamiltonian of the above optimization problem of the representative agent written as:

$$H = U(Dt, Mt) + \lambda t(rAt + t - Dt - PtMt) + \mu t(Ft - PtMt)$$

Where, in this sake $Dt$ and $Mt$ are control variables because they are included in the objective function and At as a state variable. $\lambda t$ is the co-state variable, interpreted as the marginal utility of money and $\mu t$ is the Lagrange multiplier associated with the foreign exchange constraint. Thus, for the above optimization problem we can derive the following the first order conditions:

$$\frac{dH}{dDt} = \lambda t \tag{3}$$

$$\frac{dH}{dMt} = Pt(\lambda t + \mu t) \tag{4}$$

$$\frac{d\lambda t}{dt} = (\delta - r)\lambda t \tag{5}$$

$$(Ft - PtMt) \geq 0 \text{ and } \mu t*(Ft - PtMt) = 0 \tag{6}$$

Following on the empirical frameworks [15, 18] we have assumed that (3) and (4) is an add log utility function then:

$$U(Dt, Mt) = Ct\frac{Dt^{1-a}}{1-a} + Bt\frac{Mt^{1-\gamma}}{1-\gamma}$$

Where $Ct$ and $Bt$ are random and strictly stationary shocks to preference. From the above utility function, we can derive the following first order conditions:

By inserting [15] additive log utility function into the original current value, the Hamiltonian equation is rewritten as follows:

$$L = Ct\frac{Dt^{1-a}}{1-a} + Bt\frac{Mt^{1-\gamma}}{1-\gamma} + \lambda t(rAt + t - Dt - PtMt) + \mu t(Ft - PtMt)$$

With the above function, the first order conditions rewritten as:

$$\frac{dL}{dDt} = CtDt^{-a} = \lambda t \tag{7}$$

$$\frac{dL}{dMt} = BtMt^{-\gamma} = Pt\lambda t\left(1 + \mu_t^*\right) = \lambda_t P_t^* \tag{8}$$

Where $\mu_t^* = \frac{\mu t}{\lambda t}$ is the scarcity premia, and $P_t^*$ is the scarcity price at which transactions occur at the shop floor in the secondary market. Now let's eliminate $\lambda_t$ from Eq (8) by substituting its figure in Eq (7) and if we take logarithm to get the following equation;

$$lnBt - \gamma lnMt = lnCt + lnPt - alnDt + ln(1 + \mu_t^*) \tag{9}$$

In order to derive the long -run demand for import functions, we have to impose the steady state conditions of variables $as \frac{dAt}{dt} = \frac{d\lambda t}{dt} = 0$ and as $Pt = P_t^*$. Hence total household income is, a composite of both labor and asset income which evaluated at the equilibrium price vector,

denoted by $Y_t^*$. As a result, the steady state solution implies that:

$$Y_t^* = Dt + P_t^* Mt \tag{10}$$

Using the steady state condition and taking logarithm, we get the following expression for $Kt$

$$Kt = \ln (Y_t^* - P_t^* Mt) = \ln (Yt - PtMt) \tag{11}$$

Where, $Yt = Y_t^* - \mu_t^* P_t^* Mt$ is the observed income in the regime where foreign exchange is constrained likewise $Pt.$ *is* the observed price.

Now, to eliminate $Kt$ from Eq (9) and solve for $Mt$:

$$lnMt = \frac{a}{\gamma}\ln(Yt - PtMt) - \frac{1}{\gamma}Pt - \frac{1}{\gamma}ln(1 + \mu_t^*) + \varepsilon_t \tag{12}$$

Where, $\varepsilon_t = \frac{1}{\gamma}(lnBt - lnCt)$ is a random and strictly stationary shock of preferences, $Y$ is the total expenditure by domestic consumers on both domestically produced goods and imports and the scale variable $\ln(Yt - PtMt)$ in the right hand side of equation defined as GDP minus import.

When the foreign exchange constraint is binding the Kuhn-Tucker theorem requires that $\mu t > 0$, and hence $\mu_t^* > 0$. For most of the developing countries time series data on the scarcity premia on imports, are not available. In order to make the estimating procedure easy, we need a theoretically consistent parameterization of $\mu_t^*$ in terms of the observed variables. Since $\mu_t^*$ represents the scarcity premia on foreign exchange, it should be, ceteris paribus, a negative function of the amount of foreign exchange available. So one would tend to think that a good proxy for $\mu_t^*$ can be the foreign exchange receipts ($Ft$), thus providing an ex-post rationalization of the widely used foreign exchange availability approach.

In order to capturing openness (import liberalization), through easing access to import, dummy variable has to be included in the model. Hence, forth, this dummy variable used to separate market structure categories as command economy and market oriented economy, which is important to address the effect of policy variation on import behavior.

Taking in to account the above information, we can straightforwardly recapitulate the demand for import function, which estimated with the data available in Ethiopia as follow:

$$lnMt = \frac{a}{\gamma}\ln(Yt - PtMt) - \frac{1}{\gamma}Pt - \frac{1}{\gamma}lnFt + Dm + \varepsilon_t$$

$$lnMt = \beta1 \ln(Yt - PtMt) - \beta2Pt + \beta3lnFt + \beta4Dm + \varepsilon_t \tag{13}$$

By incorporating money supply (MSt) and exchange rate volatility (VOLt) into the import demand model enriches its explanatory capacity by capturing crucial economic dynamics. The inclusion of MSt acknowledges the significant influence of monetary policy on import demand, where changes in money supply affect consumer spending and purchasing power, thereby impacting import levels. Meanwhile, VOLt reflects the uncertainty surrounding future exchange rates, influencing importers' decisions and risk management strategies. By accounting for these factors, the model not only provides a more comprehensive understanding of import behavior.

$$lnMt = \beta0 + \beta1 \ln RGDPt - \beta2Pt + \beta3lnFt + \beta4lnMSt + \beta5lnVOLt + \beta6D + \varepsilon_t \tag{14}$$

Eq (14) encompasses the determinants of import behavior, where Mt is real import, RGDPt

is real gross domestic product, Pt is relative import price, MSt is money supply, VOLt is the volatility of the exchange rate, D is a dummy variable, and $\varepsilon$t is the error term. Coefficients $\beta$0 to $\beta$6 are estimated.

### 3.2 Definition of variable and source of data

All the data are obtained from secondary sources. The data source for the study was obtained from National bank of Ethiopia (NBE), and World development indicator (WDI) from the period of 1985 to 2021.

**Dependent variable.Real aggregate imports (Mt)** are defined as the nominal values of import goods and services vary which is deflated by the import price index.

**Explanatory variables. Relative import prices (Pt)**: defined as import price index deflated by the consumer price index. It measures the responsiveness of the quantity demanded of the item that a country import to a unit price change in the imported items in domestic market.

**Real gross domestic product (Yt−PtMt)**: is an inflation-adjusted measure that reflects the value of all goods and services produced by an economy in a given year. it express based on constant (base year price). Basically, it measures the responsiveness of the quantity demanded of the item that a country import to a foreign exchange change in the income of the people who pay for it.

Foreign exchange reserves (Fx) are assets denominated in a foreign currency that are held by a nation's central bank which includes export earnings, remittances, foreign aid in the in the form of foreign marketable securities, monetary gold, special drawing rights (SDRs) and reserve position in the IMF with the intention to make international payments and hedge against exchange rate risks. Accumulation of foreign exchange reserve is used as a proxy variable to the scarcity premia on foreign exchange. It has to be expected that positively correlated with the import demand functions as the cases that when the amount of foreign exchange reserve in the central bank pocket is increasing make the nation to have more additional month coverage of import.

**Money Supply (MS)**: Supply of money broadly defined to include demand deposit and time and savings deposit. This is identically equal to the net stock of international reserve in domestic currency terms and the level of net domestic credit extended by the banking system.

**Volatility of exchange rate (VOL):** Volatile exchange rates make international trade and investment decisions more difficult because volatility increases exchange rate risk which refers to the potential to lose money because of a change in the exchange rate and has implication on the volume of international trade. It can be calculated as the standard deviation of real exchange rate between Ethiopia and its trading partners. It has expected that negatively correlated with import demand.

**Dummy variable (Dm)** is used to proxy for trade policy variation between pre-1991 (Derg regime) and post 1991(EPRDF) government on international trade and finance. For policy dummy, it takes the value 0 for the year before 1991 and takes 1 for post 1991. The trade policies of the current government have more liberalized compared to the previous government trade policies on trade in general and import in particular. Since this trade policy dummy is essential for explaining the trade policy variation.

## 4. Econometric results and discussion

The above Table 1 reveals that the logarithm of foreign exchange reserves and the volatility of the exchange rate found to be stationary at the level, significant at 1% and 10% significance levels, respectively. Conversely, the logarithm of real import, relative import price, income, and

**Table 1. Unit root test results for variable order of integration augmented dickey fuller tests.**

| Variables | ADF Test Statistics | Critical Values |
|---|---|---|
| | I(0) | I(1) |
| LnMt | -2.02 | -6.64*** |
| LnFx | -4.85*** | |
| LnM2 | -0.94 | -4.21** |
| LnRip | -1.30 | -7.30*** |
| LnY* | -2.17 | -4.29*** |
| LnVolte | -3.20* | |
| D | -6.98*** | |

(*** significant at 1% significance level

** significant at 5% significance level

* significant at 10% significance level)

Source: Own Computation from Eviews 10

money supply were not stationary at the level but became stationary after taking their first differences.

The subsequent step involves examining the existence of a long-run relationship among the variables. This is achieved by investigating if any linear combination of the series is drawn from a stationary distribution, indicating co-integration. The determination of lag length is crucial for this, and the optimal lag of one is chosen based on the Akaike information criterion (AIC) for the specified vector autoregressive model (VAR) (tested at a 5% level) in Table 2.

The pair- wise correlation matrix is adopted in this study to determine the exact relationship between the seven variables used in the study. Results from the pair- wise correlation matrix are presented in Table 3 below.

The correlation matrix in Table 3 provided offers valuable insights into the relationships among several key variables in a dataset. Each cell in the matrix contains the correlation coefficient between two variables, with values ranging from -1 to 1, indicating the strength and direction of their relationship. Notably, the natural logarithm of import demand (LNMT) exhibits a strong positive correlation with both the natural logarithm of money supply (LNM2) and foreign exchange (LNFX), implying that import demand tends to increase with higher levels of money supply and foreign exchange.

Additionally, LNMT demonstrates a moderate negative correlation with the natural logarithm of relative import price (LNRIP), suggesting a negative association between import demand and relative import prices. Conversely, LNMT displays a weak negative correlation with the natural logarithm of volatility (LNVOLT), indicating a slight tendency for import demand to decrease with increased volatility. Furthermore, LNMT shows a moderate positive

**Table 2. VAR lag order selection criteria.**

| Lag | LogL | LR | FPE | AIC | SC | HQ |
|---|---|---|---|---|---|---|
| 0 | -49.9611 | NA | 1.2849 | 3.0885 | 3.1339 | 3.1038 |
| 1 | 10.9442 | 114.4284* | 0.0340* | -0.5420* | -0.4513* | -0.5115* |
| 2 | 11.4266 | 0.8770 | 0.0351 | -0.5107 | -0.3746 | -0.4649 |
| 3 | 11.4810 | 0.0956 | 0.0372 | -0.4533 | -0.2720 | -0.3923 |

(* indicates lag order selected by the criterion)

Source: Own Computation from Eviews 10.

**Table 3. Pairwise correlation.**

|  | LNMT | LNM2 | LNFX | LNRIP | LNVOLT | LNY_ | D01 |
|---|---|---|---|---|---|---|---|
| LNMT | 1.000000 |  |  |  |  |  |  |
| LNM2 | 0.946577 | 1.000000 |  |  |  |  |  |
| LNFX | 0.782157 | 0.641774 | 1.000000 |  |  |  |  |
| LNRIP | -0.721211 | -0.647075 | -0.651487 | 1.000000 |  |  |  |
| LNVOLT | -0.327005 | -0.242189 | -0.636513 | -0.305585 | 1.000000 |  |  |
| LNY_ | 0.347420 | 0.492865 | 0.088859 | 0.681764 | 0.122543 | 1.000000 |  |
| D01 | 0.551681 | 0.474501 | 0.816274 | 0.296021 | -0.613139 | -0.137593 | 1.000000 |

Source: Own computation from Eviews10

correlation with the natural logarithm of real GDP (LNY-), suggesting a positive albeit not very strong relationship between import demand and real GDP. The correlation with a dummy variable (D01) is moderate, indicating a relationship with this variable. Similarly, LNM2 shares a strong positive correlation with LNFX and moderate negative correlations with LNRIP and a positive correlation with LNY-, suggesting similar relationships with these variables as observed with LNMT. LNFX, in turn, demonstrates a moderate negative correlation with LNRIP and weak positive correlations with LNVOLT and LNY-, indicating associations with volatility of exchange rate, and real GDP, respectively.

Moreover, LNVOLT exhibits a strong negative correlation with the dummy variable D01, implying a significant relationship between volatility and this dummy variable. Overall, this correlation matrix provides a comprehensive understanding of the interdependencies among the variables, which can be invaluable for further analysis and decision-making processes.

Moving on to the co-integration test using the bounds testing approach for the Nonlinear Autoregressive Distributed Lag (NARDL) model, the critical F-statistic values are displayed in Table 4. The F-statistic (25.181) surpasses both lower and upper bound critical values at 1%, 5%, and 10% significance levels. Consequently, the null hypothesis of no long-run relationship among the dependent variable (real import) and the other explanatory variables is rejected.

Confirming the long-run asymmetric co-integration, Table 5 illustrates the estimated long-run coefficients using the NARDL approach.

The long-run impact of the explanatory variables on real import (LNMT) is illustrated using Eq 1:

$$
\begin{aligned}
LNMT = {} & -6.6398 + 0.1666LNFX_{POS} - 0.0972LNFX_{NEG} + 1.8358LNY_{POS} + 0.3605LNY_{NEG} \\
& - 0.5284LNRIP_{POS} + 0.2904LNRIP_{NEG} + 0.0163VOLTE_{POS} - 0.0064VOLTE_{NEG} \\
& - 0.3348LNM2_{POS} + 0.4423LNM2_{NEG} - 0.7163D
\end{aligned}
\tag{1}
$$

**Table 4. Bounds test for co-integration.**

| Test Statistic | Value | Significance | I(0) | I(1) |
|---|---|---|---|---|
| F-statistic | 25.181 | 10% | 1.76 | 2.77 |
| K | 11 | 5% | 1.98 | 3.04 |
|  |  | 2.5% | 2.18 | 3.28 |
|  |  | 1% | 2.41 | 3.61 |

Source: Own computation from Eviews10

Table 5. Long run asymmetric relationship of the import demand model.

| Variable | Coefficient | Std. Error | t-Statistic | Prob. |
|---|---|---|---|---|
| LNFX_POS | 0.1666 | 0.0859 | 1.9400 | 0.0702 |
| LNFX_NEG | -0.0972 | 0.0323 | -3.0085 | 0.0083 |
| LNY__POS | 1.8358 | 0.2207 | 8.3157 | 0.0000 |
| LNY__NEG | 0.3604 | 0.4454 | 0.8092 | 0.4303 |
| LNRIP_POS | -0.5283 | 0.1013 | 5.2128 | 0.0001 |
| LNRIP_NEG | 0.2904 | 0.1532 | 1.8948 | 0.0763 |
| LNVOLT_POS | 0.0163 | 0.0029 | 5.4603 | 0.0001 |
| LNVOLT_NEG | -0.0064 | 0.0020 | -3.1953 | 0.0056 |
| LNM2_POS | -0.3347 | 0.1137 | -2.9434 | 0.0095 |
| LNM2_NEG | 0.4423 | 0.3715 | 1.1906 | 0.2512 |
| D | -0.7162 | 0.1558 | -4.5949 | 0.0003 |
| C | -6.6398 | 0.3938 | -16.857 | 0.0000 |

Source: Own Computations from Eviews 10

Based on Table 5, all the control variables exhibit an asymmetric long-run relationship with the demand for import. Foreign exchange reserve, real income, and volatility of exchange rate have a positive long-run relationship with aggregate import demand. Conversely, the policy dummy has a negative impact on import demand. The listed explanatory variables are statistically significant in explaining import demand, as indicated by absolute t-values greater than 2.

The dynamic relationships among macroeconomic indicators as determinants of aggregate import demand observed to be nonlinear. Positive and negative changes in foreign exchange reserve, relative import price, real income, and volatility of exchange rate may have differential impacts on aggregate import demand.

The NARDL estimates, presented in Table 5, provide evidence of asymmetry (nonlinear relationship) between foreign exchange reserve and Ethiopian real import. The asymmetric long-run coefficients [LNFX+ and LNFX-] indicate that both negative and positive changes in foreign exchange reserve significantly affect aggregate import demand. A negative shock in foreign exchange reserve results in a 0.09% decrease in real import, aligning with the theoretical notion that increased foreign exchange reserve enhances the capacity to import desired raw materials.

The NARDL model further highlights the asymmetric impact of control variables on real import demand. For instance, the asymmetrical relationship between real income (LNY) and aggregate import demand reveals that a negative shock in real income has no significant impact, while a positive shock increases real import by 1.83%. This aligns with the theoretical concept of the marginal propensity to import (MPM), indicating the change in imports induced by a change in income.

Relative import price also exerts a highly significant negative impact on real import demand in Ethiopia. The estimated long-run coefficients of the asymmetric ARDL model demonstrate that a positive change in relative import price significantly decreases Ethiopian real import demand. A One birr increase in relative price leads to a 0.52 birr fall in Ethiopia's demand for imports. This is in line with theoretical expectations, as higher relative import prices impede the flow of imported items.

Results from the NARDL model further provide evidence of the asymmetry in the impact of money supply (LNM2+) on real import demand. The long-run coefficients indicate that an increase in money supply negatively affects real import demand. When money supply

increases by 1%, real import demand declines by 0.33%, suggesting that an increase in money supply leads to a decrease in interest rates, causing the domestic currency to depreciate and imports to become more expensive.

Volatility of exchange rate (VOLTE) is identified as another factor influencing real import demand asymmetrically. Both positive and negative shocks in volatility significantly affect real import, with a positive change leading to a 1.6% increase and a negative change resulting in a decrease of 0.006%. The positive association between real import and volatility of exchange rate is attributed to the increased exchange rate risk associated with volatile rates, making international trade and investment decisions more challenging.

A policy dummy variable, serving as a proxy for policy variation between Derg and the current EPRDF on international trade, found to be significant and negatively related to the demand for import. The t-value of -4.59 is significant at the 1% level, suggesting that a change in trade policy regime reduces demand for import by 0.71. This aligns with economic theory, indicating that developing countries often rely on capital controls to adjust economic activities, negatively influencing the demand for import in Ethiopia.

## 4.1 Short-Term dynamics of import demand model

**Conditional error correction regression.** Using the results from Table 6, the coefficient of (ECt) is reported as -0.566. This indicates that the speed of adjustment is approximately 56.6 percent. The implication is that, in the case of a deviation from equilibrium, 56.6 percent is corrected in one year as the variable moves towards restoring equilibrium. Thus, there is strong pressure on import demand to restore long-run equilibrium whenever there is a disturbance. The speed of adjustment is statistically significant with a negative t-value of -9.74.

In the short term, a negative shock in foreign exchange reserves is found to have a positive impact on import demand in Ethiopia. Thus, aggregate imports are significantly constrained by the availability of foreign exchange in the short run. However, a positive shock of foreign exchange reserves is not significant at a 95% confidence level.

The positive shock of the lag of real income (LNY) is found to have a positive effect on aggregate import demand in Ethiopia. Due to a positive shock in real income, the demand for import increases by 1.96%. This indicates that the exogenous component of real GDP exerts a reliable, positive impact on demand for import. This study is consistent with) [19]. Conversely, a negative shock on real income does not affect real import in Ethiopia during the period under consideration.

On the flip side, a positive shock in relative import prices is also found to be statistically significant and associated directly with aggregate import demand. Due to a positive shock in relative import prices in Ethiopia, real import decreases by 0.56%. Whereas, a negative shock in relative import prices does not affect real import.

Moreover, the NARDL estimates also provide further evidence of the asymmetric relationship between the volatility of exchange rates and real import. In line with the long-run models of real import, both positive and negative shocks of the volatility of exchange rates significantly affect real import in the area under consideration. Because of a positive change in VOLTE, real import demand also increases by 0.01 percent. In addition, real import decreases by 0.006 percent, due to a negative shock in the volatility of exchange rates in the short run. It implies a positive change in volatility of exchange rates leads to demanding more import.

Additionally, the NARDL also estimates the short-run asymmetric relationship between broad money supply and real import.

**Diagnostics tests.** The Table 7 provided statistical tests offer crucial insights into the assumptions underlying the analyzed data. The White (Chi-squared) test indicates a lack of

**Table 6. Short-Term dynamics of the import demand model.**

| Variable | Coefficient | Std. Error | t-Statistic | Prob. |
|---|---|---|---|---|
| C | -7.0921 | 0.8208 | -8.6399 | 0.0000 |
| LNMT(-1)* | -1.0681 | 0.0957 | -11.1569 | 0.0000 |
| LNFX_POS(-1) | 0.1779 | 0.0945 | 1.8825 | 0.0781 |
| LNFX_NEG** | 0.1038 | 0.0361 | 2.8698 | 0.0111 |
| LNY__POS(-1)* | 1.9609 | 0.3304 | 5.9340 | 0.0000 |
| LNY__NEG(-1) | 0.3850 | 0.4732 | 0.8136 | 0.4278 |
| LNRIP_POS* | -0.5643 | 0.0993 | 5.6823 | 0.0000 |
| LNRIP_NEG | 0.3101 | 0.1599 | 1.9394 | 0.0703 |
| VOLTE_POS* | 0.0174 | 0.0033 | 5.2159 | 0.0001 |
| VOLTE_NEG* | -0.0068 | 0.0019 | -3.5739 | 0.0025 |
| LNM2_POS(-1) | -0.3576 | 0.1220 | -2.9308 | 0.0098 |
| LNM2_NEG(-1) | 0.4724 | 0.4002 | 1.1804 | 0.2551 |
| D01* | -0.7650 | 0.1878 | -4.0717 | 0.0009 |
| D(LNFX_POS) | -0.0755 | 0.0456 | -1.6557 | 0.1172 |
| D(LNY__POS) | 0.8691 | 0.4665 | 1.8629 | 0.0809 |
| D(LNY__NEG) | 0.9101 | 0.4632 | 1.9646 | 0.0671 |
| D(LNM2_POS) | -0.0135 | 0.2244 | -0.0603 | 0.9526 |
| D(LNM2_NEG) | -0.3888 | 0.4857 | -0.8004 | 0.4352 |
| ECTt-1 | -0.5662 | 0.0684 | -9.7433 | 0.0000 |

Source: Own computation from Eviews 10

NARDL Long-Run Form and Bounds Test

Dependent Variable: D(LNMT)

Selected Model: NARDL(1, 1, 0, 1, 1, 0, 0, 0, 0, 1, 1, 0)

Case 2: Restricted Constant and No Trend

Date: 04/16/22 Time: 03:40

Sample: 1985–2020

Included observations: 34

evidence for conditional heteroscedasticity, given its non-significant p-value of 0.24. Meanwhile, the Jarque-Bera test suggests that the data may adhere to a normal distribution, supported by its high p-value of 0.80. Besides, the Lagrange Multiplier (LM) test for serial correlation presents a concern with a p-value of 0.12, suggesting there is no problem of serial correlation. Additionally, the Multicollinearity test, assessed through the Variance Inflation Factor (VIF), emphasizes the absence of multicollinearity with a VIF less than 10. These tests collectively inform the validity of the statistical model.

**Table 7. Diagnostic tests.**

| Test | Null Hypothesis | T-Statistic | Probability |
|---|---|---|---|
| White (Chi-sq.) | No conditional heteroscedasticity | 20.49 | 0.24 |
| Jarque-Bera | There is a normal distribution | 2.43 | 0.80 |
| Lagrange Multiplier (LM) | No serial correlation | 2.43 | 0.12 |
| Multicollinearity(VIF) | No Multicollinearity | VIF<10 | |

Source: Own Computation from Eviews 10

## 5. Conclusion and recommendations

All the variables included in the model are integrated orders I(0) and I(1). Henceforth, the NARDL model is more appropriate. The optimum lag of the variables is one to show their asymmetric relationship. Besides, the F-statistic (25.18) is greater than both the lower and upper bound tests at 1%, 5%, and 10% significance levels. The empirical result of the NARDL model shows strong evidence of the existence of both short-run and long-run relationships among the variables included in the import demand models.

In the long run, the NARDL model result reveals that all variables are found to have a significant impact on determining aggregate import demand. For instance, the accumulation of foreign exchange reserves positively determines the demand for import. A 1% increase in foreign exchange reserves will increase real import by 3.0%. This relationship is consistent with theory and the implications of increased foreign exchange reserves. This emanates from the fact that the more foreign exchange reserves that the country has, the more capacity to import its desired raw materials.

Real income also found to have a positive significant effect on aggregate import demand in Ethiopia. This implies that import demand increases by 1.83 percent with an increase in real import by one percent.

On the contrary, relative import prices and money supply also highly affect the real import demand negatively and significantly in Ethiopia. When the money supply increases, interest rates decrease, leading the domestic currency to depreciate, making imports more expensive. Henceforth, foreign exchanges play a critical role in determining import, as they strongly affect import volume. The government should encourage local sourcing of raw materials for the productive sector, channeling these resources will bring the economy together, reduce the cost of production, and promote competitive exports that will make available foreign exchange for greater import volume. Besides, the government of Ethiopia should focus on monitoring and evaluating the quality of exportable products to overcome foreign exchange crunch. The policy implications for the government of Ethiopia must take into consideration not just one policy but a broad set of policies to assuredly determine a country's overall capacity to import.

## Supporting information

**S1 Data. Excel data used for estimation of variable.**
(XLSX)

## Author Contributions

**Conceptualization:** Mohammed Yimam Ali, Ahmed Mohammed Yimer, Tsadiku Setegn Dessie.

**Data curation:** Mohammed Yimam Ali, Ahmed Mohammed Yimer.

**Formal analysis:** Ahmed Mohammed Yimer.

**Investigation:** Tsadiku Setegn Dessie.

**Methodology:** Mohammed Yimam Ali.

**Project administration:** Mohammed Yimam Ali.

**Resources:** Mohammed Yimam Ali.

**Software:** Mohammed Yimam Ali.

**Supervision:** Mohammed Yimam Ali.

**Validation:** Mohammed Yimam Ali.

**Visualization:** Mohammed Yimam Ali.

**Writing – original draft:** Mohammed Yimam Ali.

**Writing – review & editing:** Mohammed Yimam Ali, Ahmed Mohammed Yimer, Tsadiku Setegn Dessie.

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
