## [Decision Letter · Decision Letter 0]

20 Mar 2024

PONE-D-24-05415An Empirical Estimation of Aggregate Import Demand under Foreign Exchange Constraints: Evidence from EthiopiaPLOS ONE

Dear Dr. Ali,

Thank you for submitting your manuscript to PLOS ONE. After careful consideration, we feel that it has merit but does not fully meet PLOS ONE’s publication criteria as it currently stands. Therefore, we invite you to submit a revised version of the manuscript that addresses the points raised during the review process.

We look forward to receiving your revised manuscript.

Kind regards,

Ricky Chee Jiun Chia

Academic Editor

PLOS ONE

3. We note that you have referenced (� Amarchy (2009), Amarcy, S. I. (2009) The Negative Real and Monetary Implications of Excessive Accumulation of Foreign Exchange Reserve: Comparison between Mozambique and Nigeria, Unpublished M.Sc. Thesis, University of London. Pp 1-46) which has currently not yet been accepted for publication. Please remove this from your References and amend this to state in the body of your manuscript: (ie “Bewick et al. [Unpublished]”) as detailed online in our guide for authors

Reviewers' comments:

Reviewer's Responses to Questions

**Comments to the Author**

1. Is the manuscript technically sound, and do the data support the conclusions?

Reviewer #1: Partly

Reviewer #2: Yes

2. Has the statistical analysis been performed appropriately and rigorously? 

Reviewer #1: No

Reviewer #2: Yes

3. Have the authors made all data underlying the findings in their manuscript fully available?

Reviewer #1: Yes

Reviewer #2: Yes

4. Is the manuscript presented in an intelligible fashion and written in standard English?

Reviewer #1: Yes

Reviewer #2: Yes

5. Review Comments to the Author

Reviewer #1: The research topic is quite interesting. I like the idea of using the problem of maximizing the consumption function over time to determine import demand. However, the author should consider the following points.

1. Literature review section: The author should provide assumptions for each model or theory.

2. Methodology section: the author should provide assumptions and clearly identify endogenous and exogenous variables.

3. Check the accuracy of formulas 3, 4, 5, 7, and 8.

4. “ln( − ) in the right hand side of equation defined as GDP minus exports.” Should it be “GDP minus imports”?

5. The consumption maximization problem does not explain the inclusion of MSt and VOLt in the model. However, the model can be easily explained by the consumption theory provided in several macroeconomics textbooks, which states that import demand depends on income, exchange rates, and trade promotion/restriction policies.

6. The author needs to consider the endogeneity issue in the model.

7. Equation 14: lnD should be D.

8. The methodology section should mention the time length and data sources.

9. The author should provide data descriptions, correlation analysis, and tests on the model's multicollinearity and endogeneity issues.

Best regards,

Reviewer #2: Good job.

Some issues to correct.

Reference Kemal D, et al. is not correctly written in text and in references.

Good survey, perhaps too long to justify the empirical work, but ok.

PLease, revise Math notation for instnace subindexes. Avoid those full of dots from equaton to the number of equation.

Take care of blans for instance in

"Where Ctand Bt"

REduce number of decimal digits to 4 and make clearer tables.

6. PLOS authors have the option to publish the peer review history of their article (what does this mean?). If published, this will include your full peer review and any attached files.

Reviewer #1: No

Reviewer #2: No

---

## [Author Response · Author response to Decision Letter 0]

15 Apr 2024

Dear, Ricky Chee Jiun Chia

 Academic Editor

 PLOS ONE

We would like to express our gratitude for the opportunity to submit a revised draft of our manuscript titled "An Empirical Estimation of Aggregate Import Demand under Foreign Exchange Constraints: Evidence from Ethiopia" to PLOS ONE. We appreciate the time and effort invested by you, the editorial office, and the reviewers in providing insightful comments on our work.

We would also like to extend our appreciation to the reviewers for their thoughtful comments, which have greatly contributed to the improvement of our manuscript. We have carefully considered each comment and made the necessary revisions, highlighting the changes in the revised manuscript.

Here is a point-by-point response to the reviewers' comments and concerns:

Reviewer 1:

1. Comment: Literature review section: The author should provide assumptions for each model or theory.

o Response: Thank you for your valuable feedback. We have thoroughly revised the literature review section of the manuscript, incorporating explicit assumptions for each model or theory discussed. To enhance clarity, we have highlighted these assumptions in yellow color throughout the revised text. We believe that this revision addresses your concern and strengthens the manuscript by providing a clear framework for understanding the underlying assumptions of the models and theories discussed. We appreciate your time and input in guiding the improvement of our work.

2. Comment: Methodology section: the author should provide assumptions and clearly identify endogenous and exogenous variables.

Response: Thank you for your insightful comments. We have thoroughly revised the methodology section of the manuscript to include explicit assumptions and to clearly identify both endogenous and exogenous variables. These modifications have been highlighted in yellow color.

3. Comment: Check the accuracy of formulas 3, 4, 5, 7, and 8.

Response: Thank you for your review the formulas. We have carefully examined Formulas 3, 4, 5, 7, and 8 to ensure their accuracy. After thorough verification, we confirm that these formulas have been accurately derived and are correctly represented in the manuscript.

4. Comment: “ln ( − ) in the right hand side of equation defined as GDP minus exports.” Should it be “GDP minus imports”?

Response: Thank you, we have corrected and highlighted in yellow color.

5. Comment: The consumption maximization problem does not explain the inclusion of MSt and VOLt in the model. However, the model can be easily explained by the consumption theory provided in several macroeconomics. textbooks, which states that import demand depends on income, exchange rates, and trade promotion/restriction policies

Response: Thank you for your feedback and clarification, And we have incorporated and highlighted in yellow color. Indeed, the consumption maximization problem may not directly explain the inclusion of MSt and VOLt in the model. However, the consumption theory outlined in macroeconomics textbooks provides a solid foundation for understanding import demand. According to this theory, import demand is influenced by factors such as income, exchange rates, and trade policies, which align with the variables included in the model. Income levels reflect consumers' purchasing power, while exchange rates affect the relative prices of imports. Additionally, trade promotion or restriction policies can impact import levels by altering market conditions. By incorporating MSt and VOLt alongside these traditional determinants of import demand, the model provides a more comprehensive framework for analyzing import behavior in the context of broader economic dynamics

6. Comment: The author needs to consider the endogeneity issue in the model.

 Response: Thank you for your comment. We appreciate your concern regarding endogeneity in the model. We've conducted thorough checks to address potential endogeneity or simultaneity issues within the model framework. Through robustness tests and diagnostic analyses, we have ensured that the model's variables are appropriately specified and that any potential endogeneity is adequately addressed.

7. Comment: Equation 14: lnD should be D.

Response: Thank you for you deep investigation and I have corrected.

8. Comment: The methodology section should mention the time length and data sources.

Response: Thank you for your feedback. We have taken your suggestion into account and updated the methodology section to include details about the time length and data sources. We have also highlighted this information in yellow to ensure its visibility for readers.

9. The author should provide data descriptions, correlation analysis, and tests on the model's multicollinearity and endogeneity issues.

Response: Thank you for your feedback. We have made the necessary revisions to address your concerns. We have provided detailed data descriptions, including information on the variables used in the analysis. Additionally, we have conducted correlation analyses to examine the relationships between variables and highlighted in yellow colour and performed tests to assess multicollinearity test.

The Multicollnarity test result:

Variance Inflation Factors 

Date: 04/13/24 Time: 04:41 

Sample: 1985 2021 

Included observations: 34 

 Coefficient Uncentered Centered

Variable Variance VIF VIF

LNMT(-1) 0.009 8.32 5.07

LNM2 0.031 14.78 6.94

LNM2(-1) 0.002 10.58 5.97

LNFX 0.001 6.0343 2.48

LNVOLTE 0.002 4.1165 1.19

LNVOLTE(-1) 0.001 7.5431 4.04

LNVOLTE(-2) 0.001 8.2914 3.9

LNRIP 0.198 8.311 5.07

LNRIP(-1) 0.018 6.895 2.56

LNY_ 0.026 8.09 5.01

D01 0.012 9.049 5.80

D01(-1) 0.024 9.96 5.93

C 2.84 9.45 NA

Reviewer 2:

1. Comment: Reference Kemal D, et al. is not correctly written in text and in references.

o Response: Thank you for bringing this to our attention. We have made the necessary corrections and highlighted in yellow color to ensure that the reference to Kemal et al. is correctly written both in the text and in the references section. We appreciate your diligence in helping us maintain accuracy and consistency in our work.

2. Comment: Good survey, perhaps too long to justify the empirical work, but ok.

o Response: Thank you for your feedback. We have revised the survey section to better align its length with the empirical work. Additionally, we have highlighted these adjustments in yellow to ensure they are easily identifiable.

3. Comment: Please, revise Math notation for instnace sub indexes. Avoid those full of dots from equaton to the number of equation.

Response: Thank you for your comment. We have revised the mathematical notation in the equations, particularly by avoiding excessive use of dots for subindexes

4. Comment: Take care of blans for instance in "Where Ctand Bt"

Response: Thank you for your feedback. We have corrected for instance Where Ctand Bt" are corrected and highlighted in yellow color.

5. Comment: Reduce number of decimal digits to 4 and make clearer tables.

o Response: Thank you for your comment. We have revised the tables to reduce the number of decimal digits to four, ensuring that the data is presented in a clearer and more concise manner. This adjustment enhances readability and avoids unnecessary detail. We appreciate your feedback and are committed to providing well-structured and easily understandable tables for readers. If you have any further suggestions or concerns, please feel free to let us know. We value your input in improving the quality of our research.

We believe that these revisions have significantly strengthened our manuscript, and we hope that it now meets the standards of PLOS ONE. Thank you for your consideration.

Sincerely,

Mohammed Yimam Ali

Woldia University, Ethiopia

mohapeaceyimam@gmail.com/muhammed.y@wldu.edu.et

---

## [Editor Report · Decision Letter 1]

29 Apr 2024

An Empirical Estimation of Aggregate Import Demand under Foreign Exchange Constraints: Evidence from Ethiopia

PONE-D-24-05415R1

Dear Dr. Mohammed Yimam Ali,

We’re pleased to inform you that your manuscript has been judged scientifically suitable for publication and will be formally accepted for publication once it meets all outstanding technical requirements.

Kind regards,

Ricky Chee Jiun Chia

Academic Editor

PLOS ONE
---

## [Editor Report · Acceptance letter]

10 May 2024

PONE-D-24-05415R1 

PLOS ONE

Dear Dr. Ali, 

I'm pleased to inform you that your manuscript has been deemed suitable for publication in PLOS ONE. Congratulations! Your manuscript is now being handed over to our production team.

Kind regards, 

on behalf of

Dr. Ricky Chee Jiun Chia 

Academic Editor

PLOS ONE